# Causal Discovery in Physical Systems from Videos

**Yunzhu Li**
MIT CSAIL
liyunzhu@mit.edu

**Antonio Torralba**
MIT CSAIL
torralba@csail.mit.edu

**Animashree Anandkumar**
Caltech, Nvidia
anima@caltech.edu

**Dieter Fox**
University of Washington, Nvidia
fox@cs.washington.edu

**Animesh Garg**
University of Toronto, Vector Institute, Nvidia
garg@cs.toronto.edu

## Abstract

Causal discovery is at the core of human cognition. It enables us to reason about the environment and make counterfactual predictions about unseen scenarios that can vastly differ from our previous experiences. We consider the task of causal discovery from videos in an end-to-end fashion without supervision on the ground-truth graph structure. In particular, our goal is to discover the structural dependencies among environmental and object variables: inferring the type and strength of interactions that have a causal effect on the behavior of the dynamical system. Our model consists of (a) a perception module that extracts a semantically meaningful and temporally consistent keypoint representation from images, (b) an inference module for determining the graph distribution induced by the detected keypoints, and (c) a dynamics module that can predict the future by conditioning on the inferred graph. We assume access to different configurations and environmental conditions, i.e., data from unknown interventions on the underlying system; thus, we can hope to discover the correct underlying causal graph without explicit interventions. We evaluate our method in a planar multi-body interaction environment and scenarios involving fabrics of different shapes like shirts and pants. Experiments demonstrate that our model can correctly identify the interactions from a short sequence of images and make long-term future predictions. The causal structure assumed by the model also allows it to make counterfactual predictions and extrapolate to systems of unseen interaction graphs or graphs of various sizes. Please refer to our project page for additional results: https://yunzhuli.github.io/V-CDN/.

## 1 Introduction

Causal understanding of the world around us is part of the bedrock of intelligence. This ability enables counterfactual reasoning, which often distinguishes algorithmic models from intelligent behavior in humans. This ability to discover latent causal mechanisms from data poses an important technical question towards building intelligent and interactive systems [1–3]. For instance, Figure 1 shows an example of a multi-body system. While the images may convey the identity and position of balls, the structural causal mechanism is latent. Each pair of balls is connected to each other through an edge (say a spring, a rigid rod, or be free). Further, each edge may have a set of hidden confounders, like the rest length of a spring or the rigid rod, that causally affect the physical interaction behavior. The underlying causal structure and governing functional mechanism may not be apparent if observations, such as images, are implicit measurements of ground-truth variables [4]. Furthermore, they can also vary across different configurations and scenarios within a domain. Hence, we need few-shot causal discovery algorithms purely from image data.

In a special case, where the entities are all disconnected and the only interactions are of collision-type, there have been a number of models proposed to employ an object-centric formulation in recent

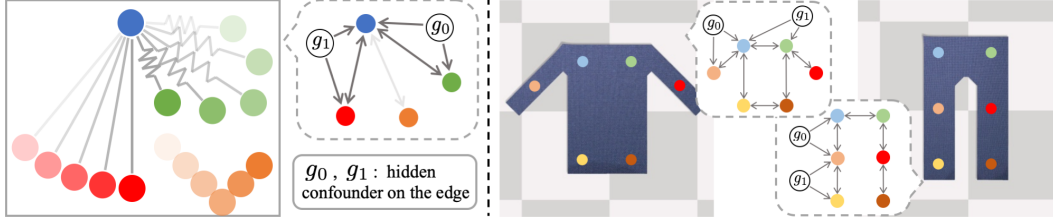

Figure 1: **Causal discovery in physical systems from videos.** The left figure shows balls, connected by invisible physical relations (shown in grey), moving around. Hidden confounding variables like edge type and edge parameters have a causal effect on the behavior of the underlying system. We humans can observe balls, infer the existence and variables on the edges between the balls, and predict the future. Similarly, in the cloth environment shown on the right, we can find a reduced-order representation by placing temporally consistent keypoints on the images and determining the causal relationships between them to reflect the cloth's topology.

literature to directly predict the future from images [5–7]. In such cases, model discovery may not even be necessary given these solutions. However, these *associative* models crumble in the face of more complex stationary underlying generative structures such as different types of latent edges and edge mechanisms [8]. Moreover, they are insufficient to capture novel generative structures and make counterfactual predictions at test time.

In this work, we aim to discover the structural causal model (SCM) to predict the future and reason over counterfactuals. To recover an SCM only from images, we need to first learn a compact state representation, infer a causal graph among these variables as well as identify hidden confounders, finally learn the functional mechanism of dynamics. This is a particularly challenging task in that we only have images and do not have explicit knowledge of the node variables. Furthermore, we neither assume access to ground truth causal graph, nor the hidden confounders and the dynamics that characterize the effect of the physical interactions. In order to tackle this end-to-end causal discovery problem in an unsupervised manner, we learn from datasets that contain episodes generated from different causal graphs but with a shared dynamics model.

**Summary of results.** The main contributions of this work lie in the one-shot discovery of *unseen* causal mechanisms in new environments from partially observed visual data in a continuous state space. This entails jointly performing model class estimation, parameter inference, and thereby building a predictive model for new latent structures at test time in a meta-learning framework.

The proposed Visual Causal Discovery Network (V-CDN) consists of three modules for visual perception, structure inference, and dynamics prediction (Figure 2). Specifically, we train the perception module that extracts unsupervised keypoints from the images to enable node discovery, building upon [9]. The inference module then takes the predicted keypoints and infers the exogenous variables that govern the interactions between each pair of keypoints using graph neural networks. Conditioned on the inferred graph, the dynamics module learns to predict the future movements of the keypoints. We consider a variety of configurations and scenarios, which gives us different combinations of variables, i.e., data from unknown interventions on the underlying system. Thus, we can hope to discover the correct underlying causal graph without explicit interventions.

Experiments show that our proposed model is robust to input noise and works well on multi-body interactions with varying degrees of complexity. Notably, our method can facilitate *counterfactual predictions* and *extrapolate* to cases with a variable number of objects and scenarios where the underlying interaction graphs are never seen before. Experiments in a fabric environment also demonstrate the generalization ability of our method, where the same model can handle fabrics of different types and shapes, accurately identifying the dependency structure and modeling the underlying dynamics even when state variables are a reduced-order keypoint-based representation of the original system.

## 2  Visual Causal Discovery in Physical Systems: V-CDN

In this section, we present the details of our model, which extracts structured representations from videos, discovers the causal relationships, infers the hidden confounding variables on the directed

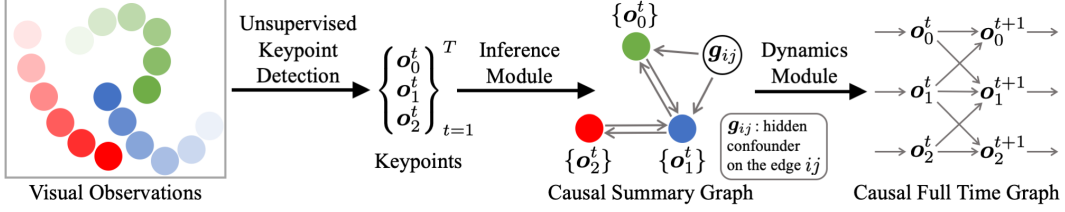

Figure 2: **Model overview.** Visual Causal Discovery Network (V-CDN) consists of three components: (a) a perception module to process the images and extract unsupervised keypoints as the state representation, (b) an inference module that observes the movements of the keypoints and determines the existence of the causal relations as well as the associated hidden confounders, and (c) a dynamics module that predicts the future by conditioning on the current state and the inferred *causal summary graph*.

edges, and then predicts the future. Our model directly learns from raw videos, which recovers the underlying causal graph without any ground truth supervision.

**Problem formulation.** We consider a dataset of $M$ trajectories observed from a latent generative dynamical system, where each datapoint is generated with unknown interventions on both the underlying causal graph structure and parameters affecting the mechanism. The generative process of each episode follows a *causal summary graph* [2], $\mathcal{G}_m = (\mathcal{V}_m^{1:T}, \mathcal{E}_m), m = 1 \dots M$, where $\mathcal{V}_m^{1:T}$ contains the subcomponents underlying the system at different time steps and $\mathcal{E}_m$, which we assume is invariant over time, denotes the causal relationships between the constituting components. Specifically, for each directed edge $(v_{m,i}, v_{m,j}) \in \mathcal{E}_m$, there are both discrete and continuous hidden confounders denoting the type and parameters of the relationship that determines the computation of the underlying structural causal model (SCM) [10] and affects the behavior of the dynamical system. We further assume that in the dynamical system, there are no instantaneous edges or edges that go back in time. Note that the *causal summary graph* may contain cycles, but when spanning over time, the derived *causal full time graph* is a directed acyclic graph (DAG), as shown in Figure 2.

In this work, we consider the case where we only have access to the data in the form of image sequences, $\mathbb{I}_m = \{I_m^{1:T}\}$, without any knowledge of the ground truth causal structure and the intervention being applied, where $I_m^t$ is an image of dimension $H \times W$, denoting the data we received at time $t$ of episode $m$. The goal is to perform a one-shot recovery of the *causal summary graph* from a short sequence of images and simultaneously learn a shared dynamics model that operates on the identified graph to make counterfactual predictions into the future. This is a particularly challenging task and our method serves as a first step for tackling this problem in an end-to-end fashion using unsupervised intermediate keypoint representations.

**Overview of Visual Causal Discovery Network (V-CDN).** We aim to find a temporally-consistent (and possibly reduced-order) keypoint-based representation from images using a perception module trained in an unsupervised way,

$$\tilde{\mathcal{V}}_m^t = f_\theta^\mathcal{V}(I_m^t), \quad t = 1, \dots, T, \tag{1}$$

where the function $f_\theta^\mathcal{V}$, parameterized by $\theta$, takes raw images as input and outputs a set of keypoints in 2-D coordinates, $\tilde{\mathcal{V}}_m^t = \{o_{m,i}^t | o_{m,i}^t \in \mathbb{R}^2\}_{i=1}^N$, that reflect the constituting components in the system. Then, we use an inference module, $f_\phi^\mathcal{E}$, parameterized by $\phi$, that takes the sequence of detected keypoints as input and predicts the edge set, $\tilde{\mathcal{E}}_m$,

$$\tilde{\mathcal{E}}_m = f_\phi^\mathcal{E}(\tilde{\mathcal{V}}_m^{1:T}), \tag{2}$$

where $\tilde{\mathcal{E}}_m = \{(o_{m,i}, o_{m,j}, g_{m,ij})\}$. $g_{m,ij}$ includes $g_{m,ij}^d$ and $g_{m,ij}^c$, denoting the latent discrete and continuous confounders associated with the directed edge from $j$ to $i$ at episode $m$. $\tilde{\mathcal{V}}_m^{1:T}$ and $\tilde{\mathcal{E}}_m$ together constitute our discovered *causal summary graph*, conditioned on which, a dynamics module, $f_\psi^\mathcal{D}$, parameterized by $\psi$, aims to predict the state of the keypoints at time $T + 1$,

$$\hat{\mathcal{V}}_m^{T+1} = f_\psi^\mathcal{D}(\tilde{\mathcal{V}}_m^{1:T}, \tilde{\mathcal{E}}_m). \tag{3}$$

By iteratively applying $f_\psi^\mathcal{D}$, we are able to make long-term future predictions.

The perception module, $f_\theta^\mathcal{V}$, the inference module, $f_\phi^\mathcal{E}$, and the dynamics modules, $f_\psi^\mathcal{D}$, are shared among all episodes in the dataset consisting of various causal graphs with different discrete and continuous hidden confounders, which enables one-shot adaptation to an unseen graph at test time and allows counterfactual predictions by intervening on the identified graph and rolling into the future using the dynamics module.

To train the system, we take an unsupervised keypoint detection algorithms [9] as our perception module and train it on the image set, $\mathbb{I}$, for extracting temporally-consistent keypoints. The inference module and the dynamics module are trained together by minimizing the following objective:

$$\min_{\phi,\psi} \sum_m \sum_t \mathcal{L}(\tilde{\mathcal{V}}_m^{t+1}, f_\psi^\mathcal{D}(\tilde{\mathcal{V}}_m^{1:t}, \tilde{\mathcal{E}}_m)) + \lambda R(\tilde{\mathcal{E}}_m), \tag{4}$$

where $R(\cdot)$ is a regularizer imposed on the identified graph, e.g., to encourage sparsity.

## 2.1 Unsupervised keypoint detection from videos

The perception module's task is to transform the images into a keypoint representation in an unsupervised way. In this work, we leverage the technique developed by Kulkarni et al. [9]. In particular, we use reconstruction loss over the pixels to encourage the keypoints to disperse over the foreground of the image. During training, it takes in a source image $I^\text{src}$ and a target image $I^\text{tgt}$ sampled from the dataset, and passes them through a feature extractor $\tilde{f}_\omega^\mathcal{V}$ and a keypoint detector $f_\theta^\mathcal{V}$. The method then uses an operation called *transport* to construct a new feature map, $\Phi(I^\text{src}, I^\text{tgt})$, using a set of local features indicated by the detected keypoints. A refiner network takes in the feature map and generates the reconstruction, $\hat{I}^\text{tgt}$. The module optimizes the parameters in the feature extractor, keypoint detector and refiner by minimizing a pixel-wise $L_2$ loss, $\mathcal{L}_\text{rec} = \|I^\text{tgt} - \hat{I}^\text{tgt}\|$, using stochastic gradient descent.

By combining the keypoint-based bottleneck layer and the downstream reconstruction task, the model extracts temporally-consistent keypoints spreading over the foreground of the images. We denote the detected keypoints at time $t$ as $\tilde{\mathcal{V}}_m^t \triangleq f_\theta^\mathcal{V}(I_m^t)$, where $\tilde{\mathcal{V}}_m^t = \{\boldsymbol{o}_{m,i}^t | \boldsymbol{o}_{m,i}^t \in \mathbb{R}^2\}_{i=1}^N$.

## 2.2 Graph neural networks as the spatial encoder

We use graph neural networks as a building block to model the interactions between different keypoints and generate object- and relation-centric embeddings. Both the inference and the dynamics modules will have the graph neural networks as a submodule to capture the underlying inductive bias.

Specifically, for a set of $N$ keypoints, we construct a directed graph $\mathcal{G} = (\mathcal{V}, \mathcal{E})$, where vertices $\mathcal{V} = \{\boldsymbol{o}_i\}$ represent the information on the keypoints and edges $\mathcal{E} = \{(\boldsymbol{o}_i, \boldsymbol{o}_j, \boldsymbol{g}_{ij})\}$ represent the directed relation pointing to $i$ from $j$, where $\boldsymbol{g}_{ij}$ denotes the associated edge attributes.

We employ a graph neural network with a similar structure as the Interaction Networks (IN) [11] as our spatial encoder, denoted as $\phi$, to generate the embeddings for the objects and the relations: $(\{\boldsymbol{h}_i\}, \{\boldsymbol{h}_{ij}\}) = \phi(\mathcal{V}, \mathcal{E})$.

## 2.3 Inferring the directed edge set of the *Causal Summary Graph*

After we obtain the keypoints from the images, we use an inference module to discover the edge set of the *causal summary graph* and infer the parameters associated with the directed edges. The inference module takes the detected keypoints over a small time window within the same episode as input and outputs a posterior distribution over the structure of the graph. More specifically, we denote the keypoint sequence as $\tilde{\mathcal{V}}_m^{1:T} = \{\boldsymbol{o}_{m,i}^{1:T}\}_{i=1}^N$. Our goal is to predict the distribution of the edge set conditioned on the keypoint sequence using the parameterized inference function, $p_\phi(\tilde{\mathcal{E}}_m | \tilde{\mathcal{V}}_m^{1:T}) \triangleq f_\phi^\mathcal{E}(\tilde{\mathcal{V}}_m^{1:T})$.

To achieve our goal, we first use a graph neural network, as discussed in Section 2.2, to propagate information spatially for each frame, which gives us both node and edge embeddings for each keypoint at each frame. We then aggregate the embeddings over the temporal dimension for each node and edge using a 1-D convolutional neural network. Another graph neural network takes in the temporal aggregations and predicts a discrete distribution over the edge types, where the first edge

type denotes "null edge". Conditioned on a sample from the discrete distribution, the model will then predict the continuous edge parameters. The edge type and edge parameters together constitute the *causal summary graph*, which determines the existence and the actual mechanism of the interactions between different constituent components.

In particular, we first propagate the information spatially by feeding the keypoints through a graph neural network $\phi^{\text{enc}}$, which gives us node and edge embeddings at each time step,

$$(\{\boldsymbol{h}_{m,i}^t\}, \{\boldsymbol{h}_{m,ij}^t\}) = \phi^{\text{enc}}(\tilde{\mathcal{V}}_m^t, \tilde{\mathcal{E}}^{\text{fc}}), \tag{5}$$

where the edge set, $\tilde{\mathcal{E}}^{\text{fc}}$, denotes a fully connected graph that contains an edge between each pair of keypoints with the edge attributes being zero. We then aggregate the information over the temporal dimension for each node and edge using 1-D convolutional neural networks (CNN):

$$\bar{\boldsymbol{h}}_{m,i} = \text{CNN}^{\text{obj}}(\boldsymbol{h}_{m,i}^{1:T}), \quad \bar{\boldsymbol{h}}_{m,ij} = \text{CNN}^{\text{rel}}(\boldsymbol{h}_{m,ij}^{1:T}), \tag{6}$$

which allows our model to handle input sequences of variable lengths.

Taking in the aggregated node and edge embeddings, we use another graph neural network, $\phi^{\text{d}}$, that only makes predictions over the edges to predict the categorical distribution over the edge type:

$$\{\boldsymbol{g}_{m,ij}^{\text{d}}\} = \phi^{\text{d}}(\bar{\mathcal{V}}_m, \tilde{\mathcal{E}}_m^{\text{d}}), \tag{7}$$

where $\bar{\mathcal{V}}_m = \{\bar{\boldsymbol{h}}_{m,i}\}_{i=1}^N$ and $\tilde{\mathcal{E}}_m^{\text{d}} = \{(\bar{\boldsymbol{h}}_{m,i}, \bar{\boldsymbol{h}}_{m,j}, \bar{\boldsymbol{h}}_{m,ij}) | 1 \le i, j \le N, i \ne j\}$. The output $\{\boldsymbol{g}_{m,ij}^{\text{d}}\}$ represents the probabilistic distribution over the type of each edge. When an edge is classified as the first type, i.e., $\boldsymbol{g}_{m,ij}^{\text{d}} = 1$ is *true*, which we denote as "null edge", it will be removed in subsequent computation and no information will pass through it. Sampling from this discrete distribution is straightforward, but we cannot backpropagate the gradients through this operation. Instead, we employ the Gumbel-Softmax [12, 13] technique, a continuous approximation of the discrete distribution, to get the biased gradients, which makes end-to-end training possible.

Conditioned on the inferred edge type $\{\boldsymbol{g}_{m,ij}^{\text{d}}\}$, we would like to predict the continuous parameter on each one of the edges. For this purpose, we construct another edge set $\tilde{\mathcal{E}}_m^{\text{c}} = \{(\bar{\boldsymbol{h}}_{m,i}, \bar{\boldsymbol{h}}_{m,j}, \bar{\boldsymbol{h}}_{m,ij}) | 1 \le i, j \le N, i \ne j, \boldsymbol{g}_{m,ij}^{\text{d}} \ne 1\}$, and use a new graph neural network, $\phi^{\text{c}}$, to predict the continuous parameters:

$$\{\boldsymbol{g}_{m,ij}^{\text{c}}\} = \phi^{\text{c}}(\bar{\mathcal{V}}_m, \tilde{\mathcal{E}}_m^{\text{c}}). \tag{8}$$

We denote the resulting edge set as $\tilde{\mathcal{E}}_m = \{(\boldsymbol{o}_{m,i}, \boldsymbol{o}_{m,j}, \boldsymbol{g}_{m,ij}) | 1 \le i, j \le N, i \ne j, \boldsymbol{g}_{m,ij}^{\text{d}} \ne 1\}$, where $\boldsymbol{g}_{m,ij} = (\boldsymbol{g}_{m,ij}^{\text{d}}, \boldsymbol{g}_{m,ij}^{\text{c}})$, indicating the topology of the *causal summary graph* with both the type and the continuous parameter of the edge effect. The inferred *causal summary graph* is then represented as $\tilde{\mathcal{G}}_m = (\tilde{\mathcal{V}}_m^{1:T}, \tilde{\mathcal{E}}_m)$.

## 2.4 Future prediction using the forward dynamics module

The dynamics module, $f_\psi^{\mathcal{D}}$, predicts the future movements of the keypoints by conditioning on the current state and the inferred causal graph: $p_\psi(\hat{\mathcal{V}}_m^{T+1} | \tilde{\mathcal{V}}_m^{1:T}, \tilde{\mathcal{E}}_m) \triangleq f_\psi^{\mathcal{D}}(\tilde{\mathcal{V}}_m^{1:T}, \tilde{\mathcal{E}}_m)$, where we instantiate $f_\psi^{\mathcal{D}}$ as a graph recurrent network, $\phi_\psi^{\text{dy}}$.

Since we are directly operating on the predicted keypoints from the perception module, the detected keypoints contain noise and introduce uncertainty on the actual locations. Hence, in practice, we represent the position in the future steps using a multivariate Gaussian distribution, where we predict both the mean and the covariance matrix of the next state for each keypoint.

## 2.5 Optimizing the model

The perception module is trained independently using the reconstruction loss, $\mathcal{L}_{\text{rec}}$. To train the inference module and the dynamics module jointly, we instantiate the objective function shown in Equation 4 by making an analogy to the ELBO objective [14]:

$$\mathcal{L} = \mathbb{E}_{p_\phi(\tilde{\mathcal{E}}_m | \tilde{\mathcal{V}}_m^{1:T})}[\log p_\psi(\hat{\mathcal{V}}_m^{T+1} | \tilde{\mathcal{V}}_m^{1:T}, \tilde{\mathcal{E}}_m)] - D_{\text{KL}}(p_\phi(\tilde{\mathcal{E}}_m | \tilde{\mathcal{V}}_m^{1:T}) \| p_\psi(\tilde{\mathcal{E}}_m)), \tag{9}$$

For the prior $p_\psi(\tilde{\mathcal{E}}_m)$, we assume that each edge is independent and use a factorized distribution over the edge types as the prior, where $p_\psi(\tilde{\mathcal{E}}_m) = \prod_{ij} p_\psi(\tilde{\mathcal{E}}_{m,ij})$. The inference module and the dynamics module are then trained end-to-end using stochastic gradient descent to maximize the objective $\mathcal{L}$.

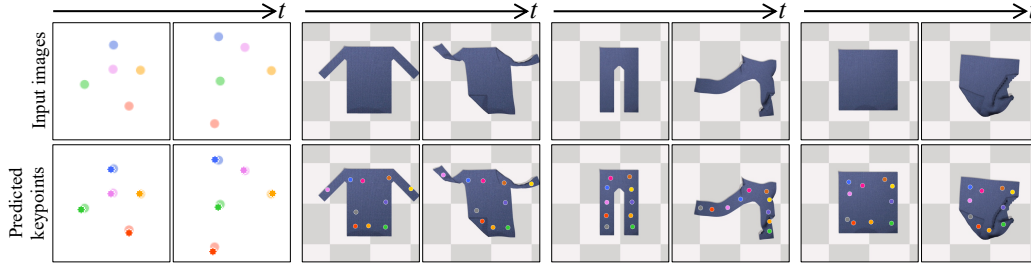

Figure 3: **Unsupervised keypoint detection.** The first row shows the input images, and the second row shows an overlay between the predicted keypoints and the image. The perception module assigns keypoints over the foreground of the images and consistently tracks the objects over time across different frames.

## 3 Experiments

The goal of our experimental evaluation is to answer the following questions: (1) Can the model perform one-shot discovery of the *causal summary graph* and identify the hidden confounders, including both discrete and continuous variables? (2) How well can the model extrapolate to graphs of different sizes that are not seen during training? (3) How well can the learned model facilitate counterfactual prediction via intervening on the identified summary graph?

**Environment.** We study our model in two environments: one includes masses, connected by invisible physical constraints, moving around in a 2-D plane, and the other one contains a fabric of various shapes where we are applying forces to deform it over time (Figure 3).

- **Multi-Body Interaction.** There are 5 balls of different colors moving around. At the beginning of each episode, we sample the invisible physical relations between each pair of balls independently, giving us the ground truth $\mathcal{E}_m$ that is fixed throughout the episode. For each pair of balls, there is a one-third probability that they are not connected or linked by a rigid rod or a spring. We also sample the continuous parameters for each existing edge and fix them within the episode, e.g., the length of the rigid relation or the rest length of the spring.

- **Fabric Manipulation.** We set up fabrics of three different types: a shirt, pants, and a towel, where we also vary the shape of the fabrics like the length of the pant leg or the height and width of the towel (Figure 5). We also apply forces on the contour of the fabric to deform and move it around. Our goal is to produce one single model that can handle fabrics of different types and shapes, instead of training separate models for each one of them.

### 3.1 Results on unsupervised keypoint detection

We employ the same architecture and training procedure described in [9] to train our perception module, $f_\theta^\mathcal{V}$. Figure 3 shows some qualitative results. Our perception module can spread the keypoints over the foreground of the image and consistently track the object. Please refer to our project page for video illustrations.

### 3.2 Discovery of the *Causal Summary Graph* and the hidden confounders

The inference module, $f_\phi^\mathcal{E}$, takes in a short sequence of the detected keypoints and aims to discover whether there is a causal relation, i.e., a physical connection, between each pair of keypoints and identifies the hidden confounders like the edge type and the edge parameters. The predicted graph will be conditioned by the dynamics module, $f_\psi^\mathcal{D}$, for future prediction. The optimization procedure does not require any supervision on the attributes associated with the edges, which allows us to infer the hidden confounders in an unsupervised way.

In the Multi-Body environment, the perception module accurately tracks the location of balls, which allows us to perform a systematic evaluation of the model's performance by comparing its prediction with the ground truth *causal summary graph* used to generate the episodes. Because we are working in an unsupervised regime, where the predicted edge type is in a discrete latent space distinguishing between null edge, spring, and rigid relation, we need to find a global one-on-one mapping between

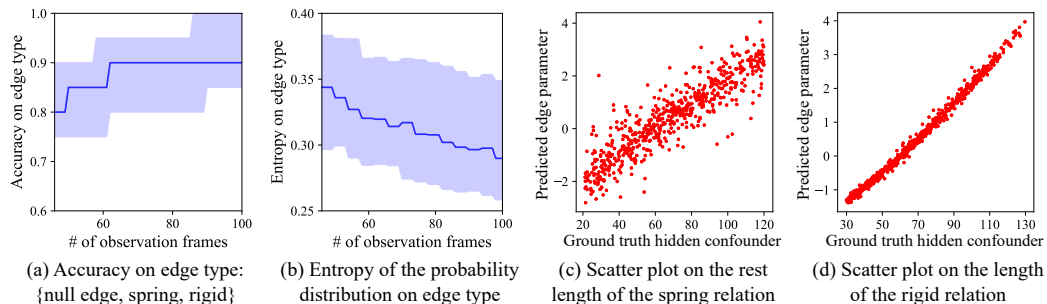

| (a) Accuracy on edge type: {null edge, spring, rigid} | (b) Entropy of the probability distribution on edge type | (c) Scatter plot on the rest length of the spring relation | (d) Scatter plot on the length of the rigid relation |

Figure 4: **Results on discovering the *Causal Summary Graph*.** Shown in (a) and (b), the accuracy of edge-type classification increases as the inference module observes more frames, which also effectively decreases the uncertainty, calculated as the entropy of the predicted distribution. As exhibited in (c) and (d), there is a strong correlation between the inferred continuous variable and the ground truth hidden confounder.

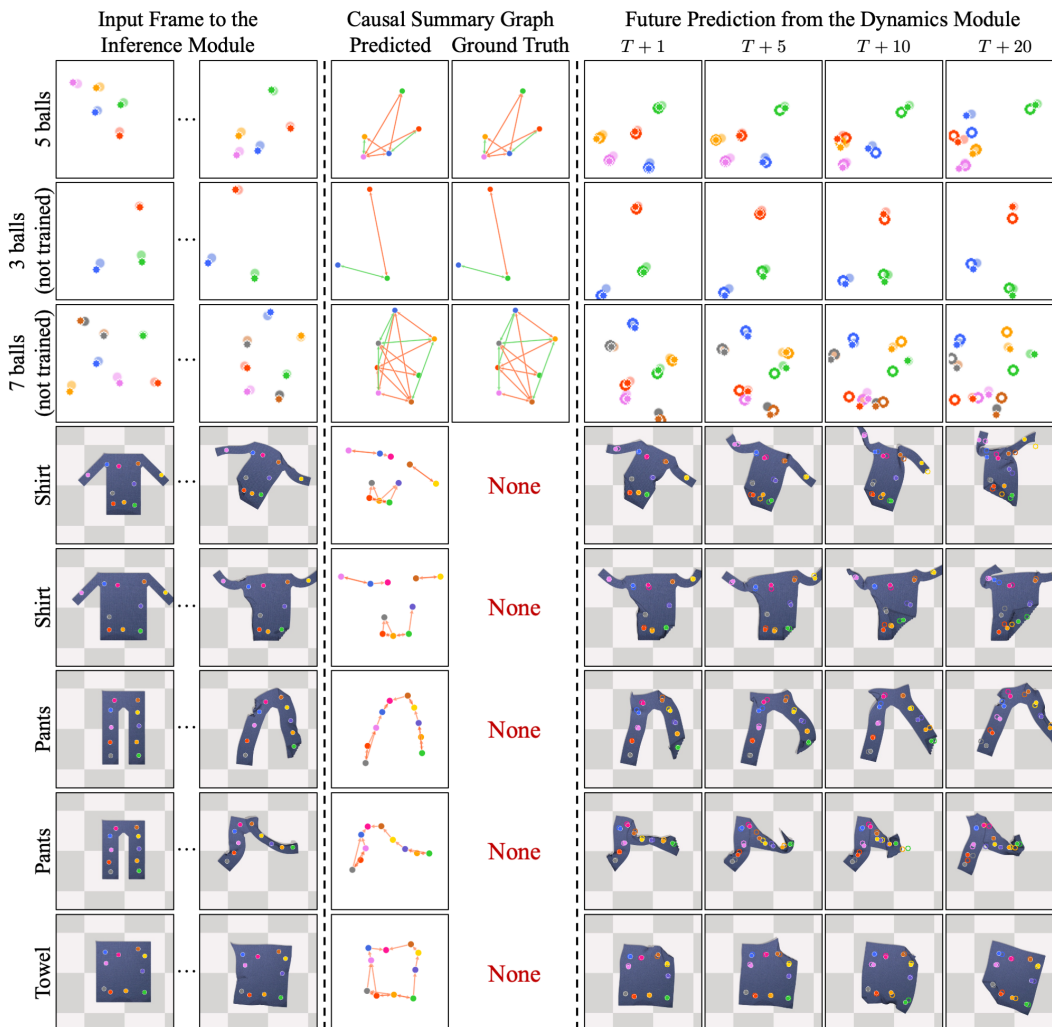

Figure 5: **Qualitative results on predicting the *Causal Summary Graph* and the future.** Our inference module observes a short sequence of images and performs one-shot discovery of the *causal summary graph*, which recovers the ground truth graph in the Multi-Body environment and captures the underlying connectivity structures in the Cloth environment. The unfilled circles in the right four columns indicate the model's prediction into the future. We overlay the predicted future keypoints with the truth future for comparison.

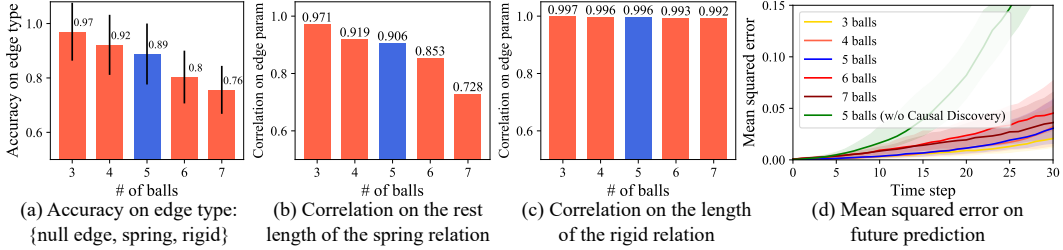

(a) Accuracy on edge type: {null edge, spring, rigid}

(b) Correlation on the rest length of the spring relation

(c) Correlation on the length of the rigid relation

(d) Mean squared error on future prediction

Figure 6: **Results on extrapolating to unseen graphs of different sizes.** Our inference module and dynamics module are trained only in environments containing 5 bodies. Thanks to the inductive bias captured by the graph neural networks in our model, it automatically generalizes to scenarios with different numbers of bodies from training. The blue bars in the figures show the performance on the test set in the same distribution we trained on, and the orange bars illustrate results on extrapolation. Surprisingly, the model has a better performance in environments with 3 and 4 balls, even if the model has never seen them before.

the prediction, $\{g_m^{\mathrm{d}}\}$, and the ground truth. We pick the one that gives us the highest accuracy, with the constraint that the first type, where there is no information passing through in the subsequent dynamics prediction, always corresponds to *null edge*. After the mapping, we evaluate the model's ability to predict the continuous confounder, $\{g_m^{\mathrm{c}}\}$, by computing its correlation with the ground truth physical parameters like rest length of the spring connection.

The results are shown in Figure 4. As the model observes more frames, the classification accuracy increases, and the uncertainty decreases, which correlates with our intuition that as we obtain more observations from the environment, we have a better estimate of the exogenous variables that govern the behavior of the system. We also show the comparison with a baseline that is the same as our method except that it does not have the inference module. Our model significantly outperforms the baseline, indicating the importance of the correct modeling of the causal mechanism (Figure 6 (d)). Figure 5 shows some qualitative results, where we include side-by-side comparisons between the identified *causal summary graph* and the ground truth.

For the cloth environment, the keypoints on the fabrics act as a reduced-order representation of the original system, where we do not know the ground truth *causal summary graph*. We encode the action as a 6-dimensional vector: the first three are the coordinates of the dragged point, and the other three indicate the movement, which will then be concatenated with the embedding of every keypoint. As shown in Figure 5, the same inference module produces different causal graphs for different types of fabrics that reflect the underlying connectivity patterns, which illustrates the model's ability to recognize the underlying dependency structure.

### 3.3 Extrapolation to unseen causal graphs of different sizes

To evaluate our model's performance on extrapolation, we also create another 4 test sets in the Multi-Body environment, including 3, 4, 6, and 7 bodies, respectively, for which we need to train separate perception modules to reflect the number of moving components. However, the inference module and the dynamics module do not require retraining; instead, they can directly generalize to systems of different numbers of bodies. As shown in Figure 6, the blue bar shows the performance on the test set that has the same number of balls as the training set, while the other bars illustrate the model's ability to perform extrapolation. Interestingly, for environments with fewer balls, e.g., 3 or 4 balls, even if the model is not directly trained on these scenarios, the performance is yet better.

### 3.4 Counterfactual prediction and extrapolation on parameter change

In our experiment, we make counterfactual predictions by intervening on the estimated hidden confounders and evaluate how well the model predicts the future by making the same intervention on the ground truth simulator. The estimated confounders are in the latent space, which requires a mapping function to get the corresponding parameters in the original simulator. We use the same mapping as described in Section 3.2 to find the corresponding discrete variables, and train a simple linear regressor for transforming the continuous variable. Figure 7 shows the performance on

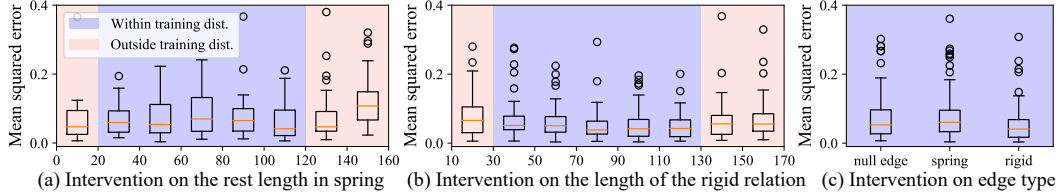

Figure 7: **Results on counterfactual prediction.** We make counterfactual predictions by intervening on the identified *causal summary graph* and evaluate the performance by comparing the predicted future with the original simulator undergoing the same intervention at $T + 30$. The modeling of the causal mechanism allows it to extrapolate to parameter ranges outside the training distribution.

counterfactual predictions, which illustrates our model's ability to answer "what if" questions and extrapolate to parameter ranges that are outside the training distribution.

## 4   Related Work

**Causal Discovery.**   Methods for causal inference from observations can broadly be categorized into three classes. Constraint-based methods (such as PC and FCI) rely on conditional independence tests as constraint-satisfaction to recover Markov-Equivalent Graphs [1, 15, 16]. Score-based methods (such as GES) assign a score to each DAG, and perform searching in this score space [17, 18]. The third class of methods exploits such asymmetries or causal footprints to uniquely identify a DAG [19–22]. Further, causal discovery from a combination of observational and interventional data has been studied in the literature [23–30]. Many of these approaches either assume full knowledge of the intervention, make strong assumptions about the model class, or have scalability limitations.

**Relational Neural Models.**   Several works have attempted modeling multi-body dynamics with graphs [7, 11, 31] and attention [32, 33]. However, these methods assume the latent generative causal graph is stationary, resulting in poor generalization to variations in either graph structure or its functional parameters. A few recent works [34, 35] have tried to infer the relationship between different entities in the system using a variational or meta-learning framework, where [34] also discussed connection to Granger causality. Still, we differ from them by directly working with image data and modeling not only the discrete but also the continuous hidden confounding variables.

**Dynamics from Videos.**   Video modeling and prediction have found much attention recently [36–39]. The idea of learned latent space embeddings for unsupervised loss computation has also enjoyed recent success in prediction [40–44]. However, the latent space may not be interpretable and overall model may not generalize. In contrast, keypoints (or particles) provide succinct and generalizable representations across a variety of use cases: particle representation [45–51], deformable object modelling [52, 53], instance independent class templates [54]. However, providing domain-specific labeled data can be tedious, hence unsupervised keypoint learning methods using reconstruction or view-consistency as loss have broader appeal [9, 55].

This paper builds on ideas from unsupervised visual representation learning and leverages it for visual causal discovery wherein the underlying model components use relational modeling to output a *Causal Summary Graph*, which has not been achieved in prior work for complex video datasets.

## 5   Conclusion

Our method extracts a structured keypoint-based representation from videos, identifies the causal relationships between different constituting components, and makes predictions into the future. The model neither assumes access to the ground truth causal graph, nor the hidden confounders, nor the dynamics that describes the effect of the physical interactions; instead, we learn to discover the dependency structures and model the causal mechanisms end-to-end from images in an unsupervised way, which we hope can facilitate future studies of more generalizable visual reasoning systems.

## Acknowledgments

We thank the entire team at NVIDIA Robotics Research Lab for their valuable feedback. We also thank the anonymous reviewers for their useful comments. The main body of this work took place when Yunzhu Li was a research intern at NVIDIA.

## Broader Impact

Causal reasoning is the process of identifying causality: the relationship between a cause and its effect, which is at the core of human intelligence. Learning directly from the observations without the modeling of the underlying causal structure can lead to the emergence of incorrect associations between the input and the output. The learned model can overfit to the bias associated with the dataset, limiting its ability to generalize outside the training distribution and often leading to catastrophic outcomes when deploying in the real world.

Discovering the causal relationships typically requires learning from data collected in randomized controlled trials or A/B tests where the experimenter controls certain variables of interest. However, carrying out the intervention or randomized trials may be impossible or at least impractical or unethical in many situations.

This work aims at discovering the causal structure and modeling the underlying causal mechanism from visual inputs, where we have access to data from different configurations and scenarios under unknown interventions both on the structure of the causal graph and its parameters. The ability to accurately capture the dependency structures and identify the hidden confounders is of vital importance towards helping the learned models generalize. As we discussed in our experiments, causal modeling improved generalization to both outside the training distribution and also towards high-likelihood counterfactual data augmentation.

While excited about these results, it is important to acknowledge that this is a particularly challenging task, and our method serves as an initial step towards the broader goal of building physically grounded visual intelligence. We mainly focus on the modeling of the dynamical system, while some aspects of the causal graph, such as sophisticated dependencies and practical issues arising from sampling rates, are not touched upon. Nonetheless, we hope to draw people's attention to this grand challenge and inspire future research on generalizable physically grounded reasoning from visual inputs without domain-specific feature engineering.

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
