[Supplementary Material]

# Causal Discovery in Physical Systems from Videos
# Supplementary Material

**Yunzhu Li**
MIT CSAIL
liyunzhu@mit.edu

**Antonio Torralba**
MIT CSAIL
torralba@csail.mit.edu

**Animashree Anandkumar**
Caltech, Nvidia
anima@caltech.edu

**Dieter Fox**
University of Washington, Nvidia
fox@cs.washington.edu

**Animesh Garg**
University of Toronto, Vector Institute, Nvidia
garg@cs.toronto.edu

## A  Additional experimental results

### A.1  Unsupervised keypoint detection

The combination of the keypoint-based bottleneck layer and the downstream reconstruction task allows the perception module to extract temporally-consistent keypoints dispersing over the images' foreground. The model accurately tracks the movement of the objects and can naturally handle deformable objects. Figure 1 shows some more qualitative examples of our perception module in both the Multi-Body and the Fabric environments.

### A.2  Future prediction in the Fabric environment

Figure 2 shows a comparison between our model and the baseline, which is the same as our model except that it does not contain an inference module to perform causal discovery. Our model can make more accurate future predictions, indicating the importance of an accurate modeling of the causal mechanisms in the underlying physical system.

## B  Model details

### B.1  Unsupervised keypoint detection from videos

The perception module maps the input images into a set of keypoints in an unsupervised way. Any unsupervised keypoint detection methods that can track the components consistently overtime should suit our use case, and there have been many recently-proposed methods that could serve this purpose [1–4]. In this work, we use the technique developed in [5].

As described in Section 2.1 of the main paper, we use reconstruction loss over the pixels to encourage the keypoints to spread over the foreground of the image. During training, it takes in a source image $I^{\text{src}}$ and a target image $I^{\text{tgt}}$ sampled from the dataset, and passes them through a feature extractor $f_\omega^{\mathcal{V}}$ and a keypoint detector $f_\theta^{\mathcal{V}}$. The model then uses an operation called *transport* to construct a new feature map using a set of local features indicated by the detected keypoints:

$$\Phi(I^{\text{src}}, I^{\text{tgt}}) \triangleq (1 - \mathcal{H}_{f_\theta^{\mathcal{V}}(I^{\text{src}})}) \cdot (1 - \mathcal{H}_{f_\theta^{\mathcal{V}}(I^{\text{tgt}})}) \cdot f_\omega^{\mathcal{V}}(I^{\text{src}}) + \mathcal{H}_{f_\theta^{\mathcal{V}}(I^{\text{tgt}})} \cdot f_\omega^{\mathcal{V}}(I^{\text{tgt}}), \tag{1}$$

where $\mathcal{H}$ is a heatmap image containing fixed-variance isotropic Gaussians around each of the $N$ points specified by $f_\theta^{\mathcal{V}}$ (Figure 1). The model then passes the feature map $\Phi(I^{\text{src}}, I^{\text{tgt}})$ through a refiner network to get the reconstruction, $\hat{I}^{\text{tgt}}$. We optimize the parameters in the feature extractor, keypoint detector and refiner by minimizing a pixel-wise $L_2$ loss, $\mathcal{L}_{\text{rec}} = \|I^{\text{tgt}} - \hat{I}^{\text{tgt}}\|$, using stochastic gradient descent.

Figure 1: **Unsupervised keypoint detection.** We show some more qualitative results of our perception module and visualize the intermediate results. In each block, the first row shows the input images, and the second row illustrates an overlay between the predicted keypoints and the image. The third and the fourth row show the intermediate results - heatmap spanned by the keypoints and the reconstructed target image.

Figure 2: **Future prediction in the Fabrics environment.** When making long-term predictions into the future, our method outperforms the baseline that does not perform causal discovery.

## B.2 Graph neural networks as the spatial encoder

Graph neural networks act as a building block in our model to capture the interactions between different keypoints and generate object- and relation-centric embeddings. Here, we describe the specific formulation of the graph neural network we used in our inference and dynamics modules.

For a set of $N$ keypoints, we construct a directed graph $\mathcal{G} = (\mathcal{V}, \mathcal{E})$, where vertices $\mathcal{V} = \{o_i\}$ represent the information on the keypoints and edges $\mathcal{E} = \{(o_i, o_j, g_{ij})\}$ represent the directed relation pointing from $j$ to $i$. $g_{ij}$ is the associated edge attributes.

Our graph neural network employs a similar structure as the Interaction Networks (IN) [6] to generate the embeddings for the objects and the relations:

$$h_{ij} = f^{\text{rel}}(o_i, o_j, g_{ij}) \qquad \text{for each edge } (o_i, o_j, g_{ij}) \in \mathcal{E}, \tag{2}$$

$$h_i = f^{\text{obj}}(o_i, \sum_{j \in \mathcal{N}_i} h_{ij}) \qquad \text{for each node } o_i \in \mathcal{V}, \tag{3}$$

where $f^{\text{obj}}$ and $f^{\text{rel}}$ are object and relation encoders respectively. $\mathcal{N}_i$ denotes all vertices that have an edge pointing to object $i$. $\{h_i\}$ and $\{h_{ij}\}$ are the derived object and relation embeddings individually. In practice, we usually propagate the node and edge information over the graph multiple times to improve the expressiveness of the model [7, 8].

The graph neural network, denoted as $\phi$, aggregates the spatial information spanned by the keypoints, passes the information along the edges, and outputs embeddings for the nodes and edges, i.e., $(\{h_i\}, \{h_{i,j}\}) = \phi(\mathcal{V}, \mathcal{E})$. Please see our main paper for how we instantiate $\phi$ as a submodule in the inference and the dynamics modules.

## C  Environment details

### C.1  Multi-Body Interaction

We use the Pymunk simulator to generate $5,000$ episodes of $500$ frames, among which $250$ episodes are reserved for testing, and the remaining goes to the training set. At the beginning of each episode, we randomly assign the balls in different positions. For each pair of balls, there is a one-third probability that they are connected by nothing, rigid rod, and spring. The stiffness of the spring relation is set to $20$, and we randomly sample the rest length between $[20, 120]$. For the rigid relation, we allow the connected two balls to move freely in a small fixed window on their opposing direction, e.g., if the rigid relationship is of length $50$, the distance between the two balls can vary between $45$ to $55$. This treatment will force the model to infer the length of the rigid relation instead of naively exploiting the distance between the two balls.

### C.2  Fabric Manipulation

We generate $2,000$ episodes of $300$ frames using the NVIDIA FleX simulator [9]. Similar to the Multi-Body environment, we reserve $200$ episodes for testing and use the remaining for training our model. As shown in Figure 1, we build fabrics of three different shapes: a shirt, pants, and a towel,

where we also vary the shape of the fabrics like the length of the pant leg or the height and width of the towel. To deform the fabrics and move them around, we apply forces on the contour of the fabric.

## D   Implementation details

Our implementation is based on PyTorch [10], and each instance of the model is trained using one NVIDIA TITAN Xp graphics card.

### D.1   Unsupervised keypoint detection

We employ a similar encoder-decoder structure as described in [5]. Both the keypoint detector, $f_\theta^\mathcal{V}$, and the feature extractor, $f_\omega^\mathcal{V}$, have 5 blocks of convolutional layers that reduce the height and width of the image into a quarter of their original size. The output of the keypoint detector has $N$ channels, representation the confidence map of the $N$ keypoints, over which $f_\theta^\mathcal{V}$ computes the exact location of each keypoint via spatial softmax. We use the operation describe in Equation 1 to get the feature maps $\Phi(I^{\mathrm{src}}, I^{\mathrm{tgt}})$. The refiner network, consisting of a few transpose convolutional operators, transforms the features map back to the original size of the target image.

We optimize $\mathcal{L}_{\mathrm{rec}}$ using Adam optimizer [11] with a learning rate of 0.001 for about 240k iterations.

### D.2   Predicting the directed edge set using the inference module

We use simple multilayer perceptron (MLP) to instantiate the object encoder, $f^{\mathrm{obj}}$, and the relation encoder, $f^{\mathrm{rel}}$. To aggregate the temporal information, we use three blocks of convolutional layers for CNN$^{\mathrm{obj}}$ and CNN$^{\mathrm{rel}}$. The use of convolutional operators allows the model to handle time series of different lengths, and the output of the CNNs is fed through a max-pooling layer to compute a fixed-dimensional feature vector.

### D.3   Joint optimization of the inference module and the dynamics module

We train the inference module, $f_\phi^\mathcal{E}$, and the dynamics module, $f_\psi^\mathcal{D}$, jointly by optimizing the loss function defined in Section 2.5 using stochastic gradient descent via Adam optimizer with a learning rate of 0.0001 in the Multi-Body environment and 0.0005 in the Fabrics environment for about 300k iterations.

For the exact network architecture and more details in the training procedures of the individual modules, please refer to our released code.