[Reviews · NeurIPS 2020]

Review 1

Summary and Contributions: The authors take on the task of causal discoveries from video data. Their goal is to "discovery the structural dependencies among environmental and object variables" and infer "the type and strength of interactions that have a causal effect on behavior of the dynamical system". Specifically, they claim contributions toward "one-shot discovery of unseen causal mechanisms in new environments from partially observed visual data in a continuous state space."

Strengths: The paper addresses ambitious goals. The evaluation attempts to examine important research questions that are clearly outlined at the beginning of Section 3. Among these questions is one about generalization ("How well can the model extrapolate to graphs of different sizes that are not seen during training?"), which is an important consideration.

Weaknesses: The ambitious goals of the introduction are not equaled by the empirical results presented later in the paper (see below). It is unclear why the authors have focused on structural causal models (SCMs), rather than some representation that is more suited to the types of models they are attempting to infer from video. Specifically, there are a variety of simulation models that are appropriate for discrete physical systems and for deformable surfaces such as cloth. These would seem a more reasonable target for learning, although SCMs provide a set of existing theory and algorithms. The authors should make clear why they focus on SCMs.

Correctness: The correctness of the authors claims are very difficult to evaluate. They provide two highly specific experimental contexts (a physical system with multiple interacting rigid objects and a deformable fabric garment). For several reasons, these results are very difficult to interpret. First, the results do not compare between the proposed systems and logical alternatives, such as impaired versions of the authors system, baselines, or alternative approaches. Second, the selected environments are few (two) and it is unclear how they were selected. Thus, readers don't have a good sense of whether these problems are particularly challenging or particularly easy. Third, the results focus on relatively esoteric measures that are relevant to the authors' methods (e.g., key point detection and discovery of the causal summary graph) rather than measures relevant to the task itself (e.g., mean-squared error over the entire surface on the cloth garments).

Clarity: The authors frequently use expansive language to describe their contributions, when that language is not supported by the specific technical work reported in the paper. For example, in the Conclusion (line 293-4), the authors state that their method "understands" causal relationships between constituting components. Better terms would be "learns", "identifies", or "constructs". Avoid using citations as nouns. For example, on line 119, the authors state that "we leverage the technique developed in [10]." It is unclear what [10] refers to until readers to consult the bibliography. Instead, say: "we leverage the technique developed by Kulkarni et al. (2019) [10]."

Relation to Prior Work: The paper discusses a set of relevant work.

Reproducibility: Yes

Additional Feedback: In their statement of broader impacts, the authors should address the potential application of their work to large-scale video surveillance, an increasing threat to individual political and social freedom throughout the world.


Review 2

Summary and Contributions: The authors describe a model for video prediction via keypoint detection and causal graph discovery, and evaluate it on image sequences from 1) a balls-in-motion domain and 2) a fabric domain. Both datasets are derived from a generative process, rather than from real images. The authors show that their method is effective, exhibits some out-of-distribution generalisation, and can carry out a form of counterfactual inference.

Strengths: The work seems to me a good development of the keypoint detection work of Kulkarni et al. (2019). Unsupervised discovery of causal structure and confounding latent variables is a difficult problem, and the paper constitutes clear progress. The evaluation is convincing, albeit in a somewhat artificial setting

Weaknesses: It would have been nice to see comparisons with more baselines. I appreciate that this is not necessarily possible with the causal graph discovery, since there is (I believe) little comparable work. But comparison would have been possible for prediction on the pixel level

Correctness: The construction of the model looks sound to me, and the evaluation seems to have been conducted well

Clarity: The paper is well written and clear

Relation to Prior Work: Yes, although I'm not sufficiently familiar with the literature to be sure nothing is missing

Reproducibility: Yes

Additional Feedback: One thing I didn't understand that could be made clearer is what actions are carried out during the generative process. In the fabric domain, from the videos supplied, it appears that parts of the fabric are pulled at random. I can't see how it's possible for the model to make predictions for many time steps into the future if this is what's happening. So maybe I'm misunderstanding something. Nothing in the supplementary material makes this clearer, so maybe the authors could explain. TYPOS Line 91: “one-short” should presumably be “one-shot” Line 133: “generates” -> “generate” Line 157: “constitutional” -> “constituent” Line 198: “one-short” -> “one-shot” Line 220: “Discovery the” -> “Discovery of the” Line 233: “subsequence” -> “subsequent”? POST-REBUTTAL COMMENTS: I have read the rebuttal and the other reviews, and it still seems like a good paper to me, so I haven't changed my score


Review 3

Summary and Contributions: This paper presents a novel framework for automatically discovering physical dynamics from videos. The proposed framework first perform an unsupervised keypoint detection and then trains GNNs for predicting the causal relations between those key points and learns the model for predicting the motion of them. The experiments on simulated datasets show that the proposed approach is capable of extracting the correct keypoints and predicting the dynamics.

Strengths: - This work introduces relational models (GNNs) to the visual dynamic prediction task; - The authors did an extensive analysis of the experimental results, which showed the strength of the presented framework; - The paper is well written, and the details of the implementation are very clear;

Weaknesses: - The presented framework is a straightforward combination of several existing state-of-the-art systems; - The key-point extraction and relational model learning are separated. As a result, the accuracy of the system highly depends on the quality of key-point detection, and it seems there is no way for the dynamic fitting module to feedback to the key-point detection module when something went wrong; - In the experiments, the proposed system is not compared with any existing baseline approaches.

Correctness: The proposed approach is solid and works on the tested tasks; the result appears to be correct.

Clarity: This paper is well written and easy to understand. The content in the supplementary is also very helpful.

Relation to Prior Work: The authors have covered a broad range of related work. However, it lacks comparisons between the presented approach to those related work in the experiment section.

Reproducibility: Yes

Additional Feedback: ---- After rebuttal ---- My major concerns about this work are lack of comparisons and the potential instability from the non-end-to-end training algorithm. In their rebuttal, the authors have included one more baseline approach and answered my second question, and I agree with them it is difficult to train an end-to-end hybrid model on such domain.


Review 4

Summary and Contributions: The authors propose a method to discover causal relationship in a simple physical system from video. The method works in a number of steps: discovering keypoints from video, inferring a causal summary graph from the a short movement sequence of these keypoints, then predicting the future movement of these keypoints. In addition to this future prediction, the model helps with counterfactual predictions. The authors claim their main contribution is one-shot discovery of unseen causal mechanisms. As I understand it, this means that they are able to detect the links/springs between masses (or between a reduced order representation of a cloth) from a few frame of video.

Strengths: The paper is easy to read and the approach to assembling different learning mechanisms (keypoint detection, graph NN) seems sound.

Weaknesses: The approach proposed in this paper seems very specific to the problem they define, namely of tracking masses that are possibly connected by rigid links or springs. As such, it provides few insights that seem transferable to other problems, even within the limited field of video understanding or prediction from video. Moreover, the simulations they use are relatively simplistic and it is unclear how well the method would work for more realistic videos of the same phenomenon.

Correctness: The proposed empirical evaluation seems hard to reproduce or extend for future works as the authors are not making video frame predictions, but rather are predicting the future position of the keypoint they extracted and which seem to correspond closely enough to the ground-truth keypoints they have available in their simulation. This fact is indeed surprising — in particular for the case of cloth simulation — given that the keypoints are extracted using the unsupervised technique described in 2.1.

Clarity: The paper is reasonably well written and easy to follow.

Relation to Prior Work: The prior work seem to be well exposed and extensive, although I do not have a deep enough knowledge of this field to validate that. However, from section 4 it was not easy for me to situate this work with respect to previous work.

Reproducibility: No

Additional Feedback: UPDATE: After reviewing the author's feedback and discussion with other reviewers, my main concern that the paper is very specific to the problem introduced here, with few generalizable learnings, still stands. My score remain unchanged.

[Author Response · NeurIPS 2020]

We thank all the reviewers for their constructive comments.

**Comparison with baselines (R1, R2, R3).**

The primary goal of this paper is to perform the one-shot discovery
of structural causal models (SCMs) from observational videos.
*Comparing Keypoint Predictions*. As R2 pointed out, there are,
unfortunately, not many baselines for us to compare. In the paper,
we included a baseline that uses a simpler prediction module using
graph-nets without causal SCM modeling. V-CDN significantly
outperforms the graph-based prediction, indicating the importance
of accurate modeling of the causal mechanism (Figure 6 (d) in the
main paper and Figure 2 in the supplement).

*Comparing Video Predictions*. Making predictions directly on a pixel level without the intermediate structures won't be
able to recover the SCM, hence lacking the ability to perform counterfactual prediction or extrapolate to unseen graphs.
Still, we follow the reviewers' suggestion by including an additional baseline that predicts directly over the pixels. We
compare with a state-of-the-art model-based reinforcement learning method, *Dreamer*[Hafner et al ICLR 2020], which
uses an encoder-decoder model to learn latent-space dynamics model from image inputs and reconstruct future image
observations. For a direct comparison, we train a visualization module that maps the predicted future keypoints from
V-CDN back to the pixel space.

The above figure shows the results. *Dreamer*'s prediction deviates from the ground truth and quickly becomes blurry,
suggesting the importance of the structured intermediate representation used in our model.

**Selection of the environments (R1, R4).** The objective of this paper is to propose a causal discovery method for
time-series data with image observations. This is a hard problem, especially with the one-shot test-time prediction setup,
which has few baselines. The method is in no way specific to these domains and the domains are chosen in order to
evaluate the generalizability of the method. The particle domain has a observational state in direct correspondence with
the ground truth node variables in the DAG, which allows us to perform a systematic analysis of the performance across
varying latent true generative models. While the fabric domain exhibits causal discovery wherein causal variables do
not trivially correspond to features in images; hence a single reduced-order model over keypoints can model different
types of fabrics. We are not aware of any other learning-based method that can predict the fabric state and its evolution
without modeling techniques that are specific to specific fabric shapes and classes. In contrast, our framework can
handle the variability of fabric structure and generalize to new shape variations and fabric parameters.

**Difficulty of the task (R1, R4).** These problems are particularly challenging because, at test time, the model has to
extrapolate to unseen edge parameter ranges (Fig. 7), edge types, and graphs of different sizes from training (Fig. 6).
Baselines, even with graph-structured prediction models, cannot cope with such out of distribution generalization.

**Applicability of the proposed method (R4, R1).** It may be a misunderstanding that our method is specific to the
problem of "tracking masses that are possibly connected by rigid links or springs" (R4). As shown in the Fabric
environment, our method can work with complicated deformable objects, where there are no explicit "masses" for
our model to track. Instead, our model extracts a structured representation and reason about the dependency structure
directly from fabric images. Meanwhile, our method does not assume to know the specific physical equations that
describe the interactions within the environment. Instead, we use graph neural networks to learn the effect of the
interactions from data and have shown in the Fabric domain that our model can model the combination of bending and
stretching force within the cloth, where writing down the analytical physical equations may be very hard as we are
operating in a reduced-order representation.

**R1: Why Structural causal models (SCMs)?** SCMs are the core of causal modeling & inference; and the underlying
generative process of the physical mechanism (often a system of differential equations) is essentially an SCM, which
corresponds to a DAG when unrolled as a causal full time graph [Peters et al.(book)] In this work, we only assume
access to visual observations. The ability to recover an SCM that closely resembles the ground truth SCM will allow the
model to perform extrapolation and make counterfactual predictions, as have been demonstrated in the paper (Figure 6
& 7).

**R2: Action carried out in the fabric domain.** We apply random forces on the contour of the fabric to deform and
move it around, where we encode the action as a 6-dimensional vector: the first three is the coordinate of the dragged
point, and the other three indicate the movement, which will then be concatenated with the embedding of every keypoint.

**R3: Separation between key-point extraction and relational model learning.** We have tried to train the modules
jointly in an end-to-end framework. However, due to the interplay between many competing losses, we observed
degradation in the perception module and degeneracy of the detected keypoints. Still, this is an open question for future
work on better architectures for end-to-end training.

We will release code to support reproducibility and also correct the typesetting, citation format and adjust the language
to be precise. We will also add discussion in broader impact of V-CDN in camera-ready version.

[Meta-Review · NeurIPS 2020]

This paper has received 4 reviews, with two reviewers recommending acceptance and two reviewers recommending rejection. The AC had ensured that the reviewers pool was consistent with the mix of topics covered by the paper, namely machine learning, computer vision, and causal inference. Given the mixed reviews and continuing disagreement between reviewers after the end of the discussion phase, the paper has received particular attention and was discussed between an AC and a Senior AC. Several strengths were identified:(i) unsupervised estimation of a structured causal model from visual data; (ii) a novel unsupervised estimation of causal graphs (a structural causal model) based on prior work [10] on unsupervised keypoint estimation combined with graph networks; (iii) evaluation on two different types of dynamical data, including classical mass-sprint systems and soft fabrics. GT SCMs are available for the former, but not the latter. The reviewers and ACs also identified weaknesses, which were mostly of experimental nature: (i) The main paper lacks crucial information on the processes generating the data, and while some more information is given in the supplementary material, this description is still not sufficient, in particular in the case of the fabrics dataset; (ii) the data is of limited complexity, the underlying hyper-parameters are limited to the number of balls in the mass-spring system.(iii) two reviewers critisized lack of generalization (this point has been addressed by the rebuttal, but did not convince some of the reviewers). The ACs concur that generalization is not tested in the fabrics scenario (e.g. beyond fabric types), but do not agree on the very general points on generalization raised by some reviewers. After discussion, the AC and SAC judged that experimental validation was sufficient and that its limitations were outweighed by the novelty of the method. While they acknowledge the validity of the points raised by the two more critical reviewers, they also judge that the paper is partially motivated by the estimation of an SCM from visual data (as also pointed out in the rebuttal) and not just by the resolution of a particular problem in ML, and recommend acceptance.